**communications** engineering

# Reliable and efficient transcranial magnetic stimulation–electroencephalography (TMS–EEG) using ultra-thin active electrodes
Johannes Gruenwald [1] ✉, Leonhard Schreiner [1,2], Sebastian Sieghartsleitner [1,3], Alexandru Buzamat[4], Giovanni Lombardi[1], Antonio Calzone[1], Marco Fummo[1], Slobodan Tanackovic[1], Marian-Silviu Poboroniuc[4], Rossella Spataro[5], Agnese Zazio[6], Marta Bortoletto[7] & Christoph Guger[1]

Transcranial magnetic stimulation combined with electroencephalography enables direct assessment of cortical excitability and connectivity via stimulation-evoked responses. While passive electrodes remain the gold standard, they require extensive preparation and are sensitive to contact quality, whereas active electrodes are easier to use but limited by increased height and larger decay artifacts. Here we introduce an ultra-thin active electrode system combined with hardware- and software-based artifact suppression. We collected electroencephalographic data from 10 healthy adults in Austria in 2024, recording brain responses with both active and passive electrodes during stimulation of the left primary motor cortex. We analyzed early and late responses for similarity and amplitude variability and found high consistency between electrode types, with stable waveforms after 20–30 trials. Response amplitudes did not differ significantly between electrode types. These findings demonstrate that active electrode systems can provide reliable and efficient recordings, supporting their broader use in stimulation–electroencephalography research.

Transcranial magnetic stimulation (TMS) is a non-invasive technique that uses magnetic fields to induce electrical currents in specific brain regions, enabling researchers to map brain function and explore the excitability of different regions[1]. When combined with electroencephalography (EEG), TMS–EEG provides a powerful tool for directly measuring electrophysiological brain responses to TMS, offering insights into the temporal dynamics of cortical excitability and network interactions[2–4].

The phase-locked responses elicited by TMS, known as TMS-evoked potentials (TEPs), reflect the functional state of the brain and its response to external perturbation in real time[5–7]. They exhibit a characteristic morphology composed of negative and positive peaks at specific latencies that originate both from the directly stimulated cortical site and from secondary activations in distant regions[8–10]. For example, after primary motor cortex (M1) stimulation, early components include both a positive and a negative peak around 15 ms post-stimulation (P15 and N15, respectively) and a positive peak around 30 ms (P30);

subsequent components, such as the N45, P60, N100, and P180 that are thought to reflect progressively later stages of signal propagation in cortical networks[10–14].

Since the early integration of TMS and EEG, numerous studies have demonstrated the utility of TEPs in research on neuropsychiatric disorders, as major depression and schizophrenia[15–19]. Moreover, TEPs have been shown to provide valuable insights into network-level neurophysiology in conditions such as sleep disorders[20], systemic diseases[21], as well as stroke[22].

Passive Ag/AgCl electrodes are a widely used and well-established technology for EEG recording and have also become the de-facto standard for TMS–EEG applications[2,23]. Despite their broad adoption and lack of viable alternatives, passive electrodes present several challenges in the context of TMS–EEG. Most notably, they are highly susceptible to TMS-induced artifacts, including large decay artifacts and amplifier saturation, particularly during the first milliseconds following stimulation[2,24–26]. Minimizing these artifacts requires maintaining homogeneous electrode

¹g.tec medical engineering GmbH, Schiedlberg, Austria. ²Institute for Integrated Circuits, Johannes Kepler University, Linz, Austria. ³Institute of Computational Perception, Johannes Kepler University, Linz, Austria. ⁴Faculty of Electrical&Power Engineering and Applied Informatics, Gheorghe Asachi Technical University, Iaşi, Romania. ⁵Intensive Neurorehabilitation Unit, Villa delle Ginestre Hospital, A.S.P. Palermo, Palermo, Italy. ⁶Neurophysiology Lab, IRCCS Istituto Centro San Giovanni di Dio Fatebenefratelli, Brescia, Italy. ⁷MoMiLab, IMT School for Advanced Studies Lucca, Lucca, Italy. ✉e-mail: gruenwald@gtec.at

impedances below 5 kΩ, which in turn requires extensive scalp preparation like skin abrasion followed by standard gel injection. As a result, setup times are substantial, especially in high-density configurations.

To address these limitations, recent work has explored the use of active EEG electrodes in TMS–EEG setups[27,28]. These electrodes incorporate integrated preamplifiers located close to the scalp, which enhance signal quality by reducing cable and movement artifacts. In addition, they tolerate higher and more variable impedances (up to 50 kΩ), enabling faster setup by reducing the need for extensive skin preparation.

While active electrodes offer clear advantages, several limitations have prevented their widespread adoption in TMS–EEG research. A primary issue is the increased coil-to-scalp distance resulting from the considerable height of many active electrodes (e.g., ≈6 mm), which is due to the integrated preamplification circuitry. This increased distance requires higher stimulation intensities[27,29], leading to louder coil clicks and stronger auditory artifacts, while also reducing stimulation focality by enlarging the volume of cortical tissue activated by the magnetic field. Furthermore, the combination of preamplification circuitry, higher stimulation intensities, and higher electrode impedances can result in more pronounced and prolonged TMS-induced decay artifacts that may obscure early TEP components[29–31]. Finally, most existing artifact removal approaches remain non-standardized and depend on manual or expert-driven procedures, limiting their straightforward applicability[25,26,32–34].

In this work, we address these issues using a novel TMS–EEG setup that integrates ultra-thin active electrodes with a height (only 3 mm) comparable to state-of-the-art passive electrodes. The system includes a dedicated TMS electrode connector box with integrated electronics that attenuate TMS-induced decay artifacts at the hardware level. Additionally, we employ a fully automated algorithm that reliably eliminates residual decay artifacts, enabling fast and standardized post-processing without the need for expert intervention.

To validate our approach, we conducted a systematic comparison between our active TMS–EEG setup and a gold-standard passive configuration. Using simultaneous recordings from active and passive electrodes in 10 healthy subjects stimulated over the left M1, we evaluated signal consistency within and between electrode types, modeled TEP amplitude variability using a linear mixed-effects framework, and analyzed TEP convergence as a function of trial count. By examining both early (15–80 ms) and late (80–350 ms) TEP components, we demonstrate the feasibility, reliability, and efficiency of using active electrodes in TMS–EEG recordings.

## Results

Figure 1 presents a topographical overview of TEPs, shown as averages across 100 trials (excluding rejected trials) for subjects S1 to S10. All electrodes (except FCC5h in subject S6) show clean and plausible TEPs, including typical components (P30, N45, P60) with amplitudes decreasing with distance from the stimulation site C1. Notably, subjects S1–S3 (and, to a lesser extent, also S7) exhibit distinct responses immediately following the stimulation pulse. We address this observation in the Discussion section.

In the following subsections, we quantitatively assess signal consistency within and between electrode types, examine factors contributing to TEP amplitude variability, and evaluate the convergence of TEP averages toward the full-trial average.

### Decay artifact removal

Figure 2 shows the performance of the decay artifact removal procedure for a passive (C2) and an active (FCC2) electrode in four representative subjects (S3, S5, S6, and S7). We selected these subjects to illustrate the range of signal characteristics and decay artifact profiles observed across the dataset.

For each subject and electrode, we fit a decay artifact model, consisting of superimposed exponential decay functions, to the original, epoched, and averaged signals (upper plots; see Methods for details). We then subtracted the decay artifact fit from the original signal to obtain the cleaned TEPs

(lower plots). Notably, the original signals show substantial variability across subjects and electrodes.

In S3, the passive electrode displays a prototypical exponential decay characterized by a single dominant component, while the active electrode reveals a more complex waveform with alternating slopes and polarities, indicating multiple decay components with various time constants. Immediate post-stimulation components (<10 ms) are visible in both S3 and S7 and remain preserved after artifact removal. S5 and S7 exhibit late responses (N100, P200), which are potentially modulated by the auditory click. Asymmetric decay dynamics are illustrated in S6: the passive electrode settles quickly, whereas the active electrode exhibits a more pronounced and prolonged decay.

Despite these variations, and regardless of the underlying artifact composition, polarity, or time constants, the decay artifact removal procedure preserves all physiological components – from immediate to late responses – yielding nearly identical TEPs for both active and passive electrodes.

### Signal Consistency

Figure 3 presents four quantitative analyses of signal consistency, comparing active and passive electrode types across various neighboring electrode configurations. In Fig. 3a, we analyzed all available neighboring electrode pairs across both electrode types, resulting in 15 pairs per subject and a total of N = 150 pairs, each separated by an approximate distance of Δ ≈ 2.1 cm. For early TEPs (15–80 ms post-stimulus), we observed a median concordance correlation coefficient (CCC) of 0.97 with an interquartile range (IQR) of [0.94, 0.99]. For late TEPs (80–350 ms), the median CCC was 0.96 [0.92, 0.99].

In Fig. 3b, we assessed signal consistency within passive electrodes by analyzing diagonal neighbors (N = 100, Δ ≈ 3.0 cm), yielding a CCC of 0.96 [0.91, 0.98] for early and 0.95 [0.92, 0.97] for late TEPs. Figure 3c shows the analysis of horizontal neighbors within passive electrodes (N = 50, Δ ≈ 4.2 cm), resulting in a CCC of 0.92 [0.79, 0.96] for early TEPs and 0.93 [0.88, 0.95] for late TEPs.

Finally, due to the topography of the active electrode layout, only horizontal neighbor comparisons were feasible (Fig. 3d). Here, the CCC was 0.92 [0.86, 0.95] for early TEPs and 0.90 [0.85, 0.94] for late TEPs.

Filled circles in the early TEP signal consistency plots indicate outliers with CCC < 0.5 that fall below the axis limits (active vs. passive (a): 2 outliers; within passive, diagonal (b): 1 outlier; within passive, horizontal (c): 1 outlier; within active (d): 1 outlier).

### Amplitude variability

To investigate the effects of electrode type and stimulation distance on TEPs, we analyzed TEP amplitudes (root mean square, RMS) for early (15–80 ms) and late (80–350 ms) components using linear mixed-effects models. In both time windows, TEP amplitude decreased significantly with increasing distance from the stimulation site (early: β = –0.136, p < 0.001; late: β = –0.077, p < 0.001). Electrode type (active vs. passive) had no significant effect (early: β = –0.031, p = 0.62; late: β = –0.076, p = 0.16). The models revealed substantial between-subject variability in overall TEP amplitude (intercept standard deviation (SD): 0.83 early, 1.61 late), and in the effects of distance (SD: 0.07 early, 0.06 late) and electrode type (SD: 0.06 early, 0.06 late). Residual variability was relatively low (SD: 0.32 early, 0.26 late), indicating that the inclusion of random slopes captured meaningful inter-individual differences in how electrode type and distance influenced TEPs.

Figure 4 provides a qualitative visualization of the TEP amplitudes across the anterior row of the electrode montage. A clear amplitude gradient with increasing distance from the stimulation site (C1) is visible for both early and late components. For clarity, we normalized amplitudes for each subject by the average RMS across all channels.

### Convergence of averages

Figure 5 shows the convergence of averages for early and late TEPs toward the full-trial average across all available trials. Active and passive electrodes

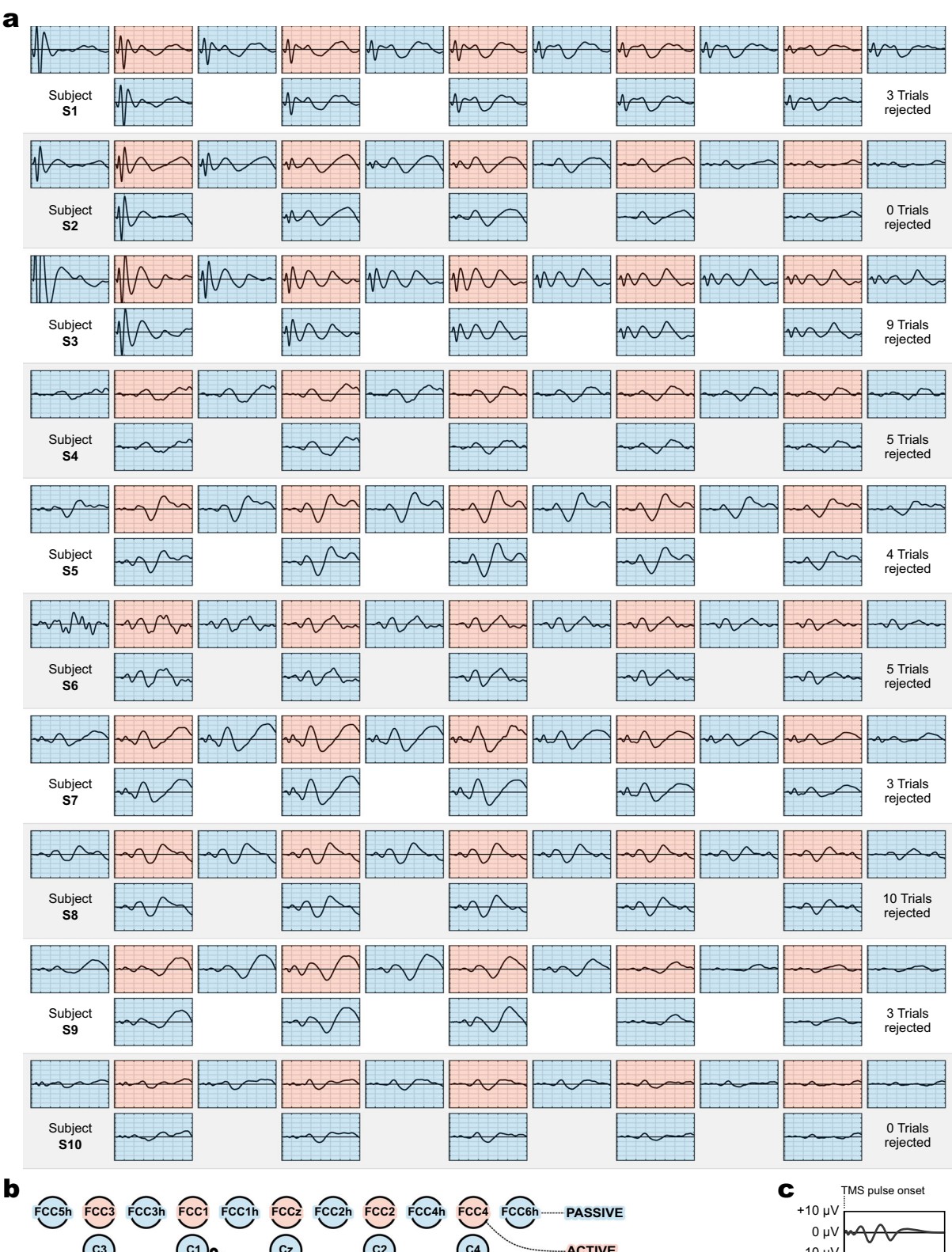

**Fig. 1 | Topographical distribution of TMS-evoked potentials (TEPs) across subjects. a** Subject-specific visualization (S1–S10). **b** Electrode montage. **c** Axes. Light blue and red shading indicate passive and active electrode type, respectively. Coil location was above C1 as indicated.

exhibit nearly identical convergence behavior. Early TEPs converge more rapidly, reaching a median CCC of 0.8 after 20 trials, whereas approximately 40 trials are needed for late TEPs. Likewise, a median CCC of 0.9 is achieved after 30 trials for early TEPs, but it takes around 60 trials to reach the same level for late TEPs.

## Discussion

In this work, we addressed key challenges associated with the use of active EEG electrodes in TMS–EEG recordings. The ultra-thin active electrodes presented in this study – measuring 3 mm in height – are now on par with passive gold-standard electrodes. This results in coil-to-scalp distances

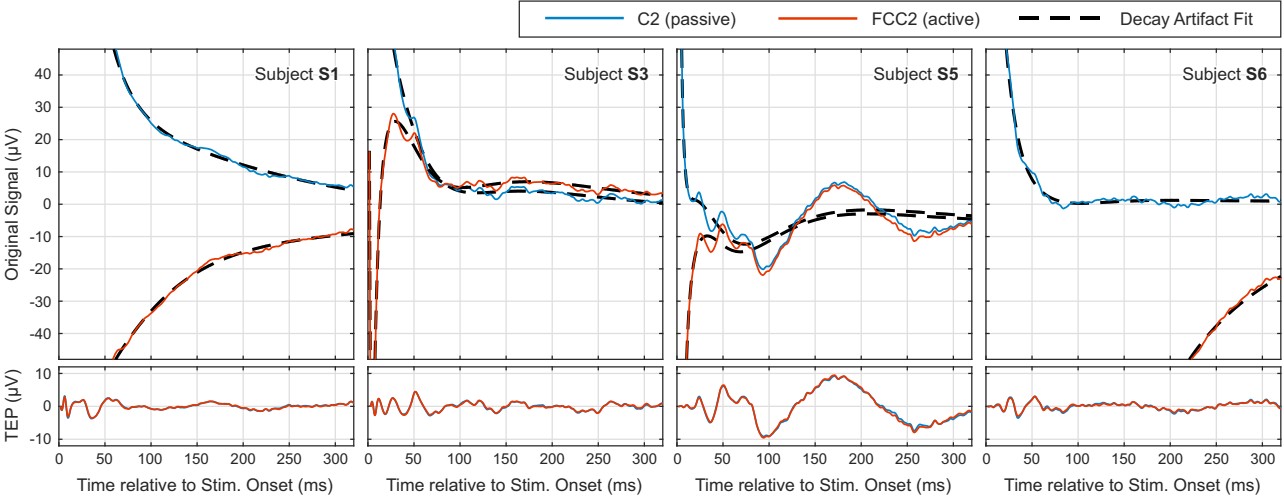

**Fig. 2 | Examples of decay artifact removal performance.** Upper plots: Original epoched and averaged signals with the decay artifact fit. Lower plots: Resulting TMS-evoked potentials (TEPs) after artifact removal. Note the identical amplitude scale across all plots.

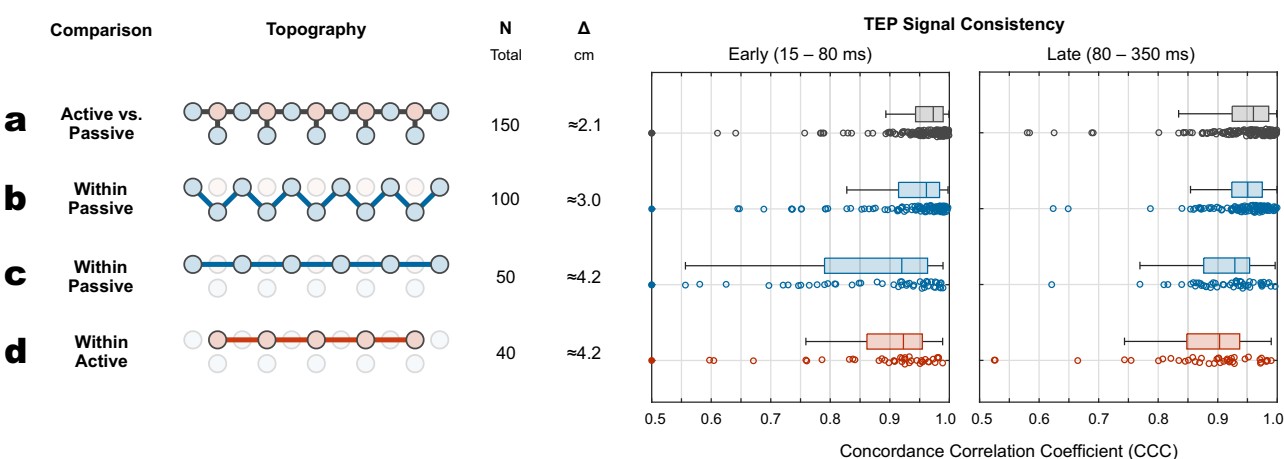

**Fig. 3 | Signal consistency analysis across different electrode configurations.** **a** Active vs. passive electrodes. **b** Within passive electrodes – diagonal pairs. **c** Within passive electrodes – horizontal pairs. **d** Within active electrodes – horizontal pairs. Light blue and red shading indicate passive and active electrode type, respectively. N: total number of electrode pairs across all subjects. Δ: approximate distance between electrode pairs. Filled circles in the TEP signal consistency plots indicate outliers beyond the axis limits.

comparable to passive setups, enabling lower stimulation intensities and improved focality relative to conventional active systems. Our setup also mitigates the problem of strong or prolonged TMS decay artifacts, commonly observed in active TMS–EEG, by combining a dedicated TMS electrode connector box with integrated circuitry for hardware-level artifact attenuation and an automated artifact removal software algorithm. This dual approach yields TEPs that are virtually free from decay artifacts, allowing clean analysis of both early and late components. Notably, artifact removal in the proposed setup is entirely automated, eliminating the need for manual intervention. As a result, the system enables reproducible, scalable analyses and lowers the barrier to adopting active electrodes in TMS–EEG research

## Active TMS–EEG is reliable

As shown in Fig. 1, TEPs across all channels and subjects exhibited stable and interpretable waveforms, except for a single noisy channel (S6, FCC6h). From a quantitative perspective, TEPs were consistent both within and between electrode types, as illustrated in Fig. 3. Here, CCC values were uniformly high across neighboring electrodes, averaging between 0.90 and 0.97 depending on the specific comparison. Moreover, CCC values

decreased predictably with increasing distance from the stimulation site (Fig. 4), reflecting a plausible physiological gradient. This finding aligns with recommendations that physiologically meaningful TEPs should exhibit stronger responses near the stimulation site and weaker responses at greater distances[29]. This spatial pattern supports the physiological validity of the recorded TEPs and is consistent with prior studies showing that TEPs are highly sensitive to changes in stimulation parameters (e.g., intensity, coil location and angle), while remaining stable and reproducible when these parameters are held constant[35–37].

Using a linear mixed-effects model, we further confirmed that electrode type had no significant effect on TEP amplitude, indicating that active and passive electrodes yield comparable results. Similar conclusions have been reported, showing that both electrode types produce highly similar TEP amplitudes and spatial patterns across various stimulation sites[27]. These findings, along with ours, reinforce the notion that the choice between active and passive systems can be guided by practical considerations without compromising data quality.

Finally, our convergence analysis demonstrated the robustness of TEP averaging: CCC values systematically increased toward the full-trial average

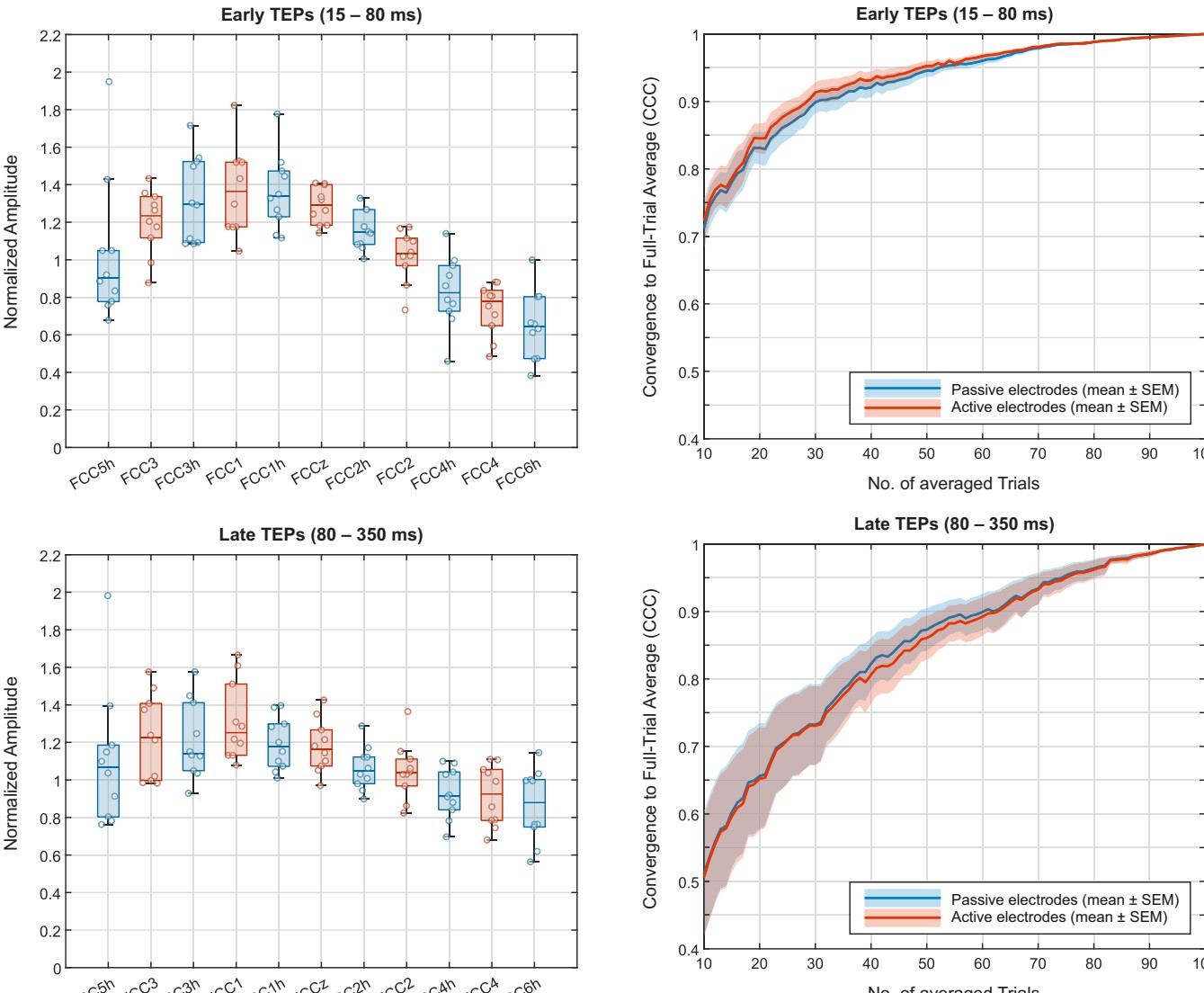

**Fig. 4 | Lateral progression of normalized TMS-evoked potential (TEP) amplitudes across the anterior electrode row.** Values are shown from leftmost (FCC5h) to rightmost (FCC6h) electrode. Blue: passive electrodes; red: active electrodes.

**Fig. 5 | Convergence to the full-trial TMS-evoked potential (TEP) average.** The solid line represents the mean, and the shaded area indicates the standard error of the mean (SEM). Passive and active electrodes are shown in blue and red, respectively.

as more trials were accumulated. This is particularly relevant, as trials contaminated by external artifacts (e.g., subject movement) can otherwise hinder convergence – a phenomenon we did not observe to a substantial extent, likely due to the effectiveness of the implemented trial rejection procedure. Previous studies have shown that the stimulation site critically influences the minimum number of pulses required for stable TEPs[38]. Specifically, stable TEPs from the primary motor cortex often stabilize with fewer than 100 pulses[27,28,39], whereas parietal cortex TEPs may require between 130–180 pulses[40–42]. As shown in our convergence analysis (Fig. 5), only a subset of the final 100 trials was required to reliably approximate the TEP. For instance, a median CCC of 0.9 was reached after 30 trials for early TEPs.

Taken together, these findings support the reliability of the proposed active TMS–EEG setup.

### Active TMS–EEG is efficient

The proposed active setup also offers substantial gains in experimental efficiency. Preparation time was considerably reduced: while passive electrodes required careful skin abrasion and impedance control below 5 kΩ, active electrodes performed reliably at impedances up to 20–30 kΩ, enabling much faster setup. All seven active electrodes could be prepared

within 1–2 minutes, compared to 20–30 minutes for the 13 passive electrodes. This observation was consistent across all subjects, suggesting that active electrodes can be prepared at least five times faster than passive ones. The increased efficiency in EEG preparation will particularly beneficial for both clinical research and practice, where stringent time constraints often limit the feasibility of high-density TMS–EEG recordings.

A key contributor to the efficiency of the proposed active TMS–EEG setup is the low-profile, flat-surface design of the active electrodes. This design enables more stable and reproducible coil placement and minimizes the coil-to-scalp distance, which in turn reduces the required stimulation intensity, stimulation-related artifacts, and subject discomfort. Supporting this, even small increases in coil–cortex distance have been shown to substantially elevate motor thresholds, underscoring the importance of maintaining minimal distance for effective and comfortable stimulation[43].

The implemented trial rejection procedure further enhances efficiency by automatically removing trials contaminated by external artifacts. Automated trial rejection has previously shown to improve data quality, enhance the reliability and reproducibility of TMS–EEG measurements, and reduce manual preprocessing workload[44].

**Table 1 | Overview of electrode types used in this study**

|  | Passive Electrode | Active Electrode |
|---|---|---|
| Product name | Multitrode Electrode B18 for TMS | g.LADYbird TMS |
| Manufacturer | Easycap GmbH, Wörthsee, Germany | g.tec medical engineering GmbH, Schiedlberg, Austria |
| Pre-Amplification | no | yes |
| Interface Material | Sintered Ag/AgCl | Sintered Ag/AgCl |
| Interface shape | Intermittent ring | pellet |
| Height (mm) | 3.5 | 3.0 |
| Diameter (mm) | 14 | 18 |

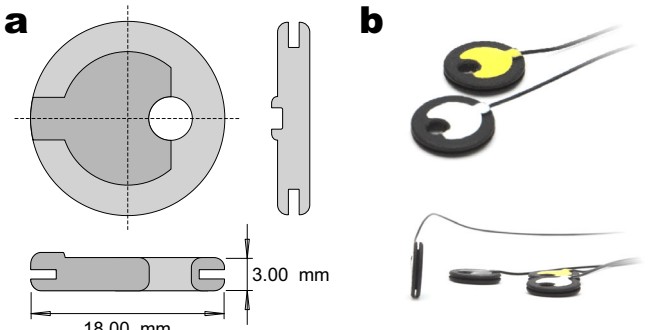

**Fig. 6 | Active electrodes used in this study. a** Technical drawing. **b** Photograph.

Likewise, our implemented artifact removal contributes to workflow efficiency by allowing rapid, consistent data cleaning without requiring manual intervention. This is particularly advantageous in TMS–EEG studies, where large datasets require efficient preprocessing pipelines. Automated artifact rejection methods can substantially reduce preprocessing time while maintaining high data quality, thereby enabling real-time or near-real-time data analysis in both experimental and clinical settings[45]. Although our artifact removal was applied offline in the current study, the algorithm is real-time capable by design. Performance benchmarks indicate a typical processing time of 150-200 ms per channel, without specific performance optimizations.

### Further considerations and limitations

In all analyses, we used the active reference for consistency across electrodes. As a control, we repeated the key analyses using the passive reference, which yielded nearly identical results (see Supplementary Discussion 1–5). This supports the robustness of our findings against referencing scheme differences.

Active electrodes are typically associated with stronger decay artifacts than passive electrodes, which can lead to amplifier saturation following the TMS pulse. However, in our experiments, we did not observe such saturation: signals from as early as 1.5 ms after stimulation onset were preserved and usable. Interestingly, we observed immediate responses (<10 ms) that appeared to be of physiological origin (see S1–S3 and S7 in Fig. 1)[46,47]. These components were not removed by the decay artifact removal algorithm, which selectively compensates for stereotyped exponential discharge patterns but not for oscillatory activity.

Although these immediate responses were not anticipated and were therefore not part of our experimental focus, their presence raises interesting questions about the feasibility of using active TMS–EEG for capturing ultra-early cortical or peripheral activity following TMS. We did not include these immediate responses in our quantitative analyses because they are highly focal and not present in all subjects and channels, which makes them unsuitable for signal consistency metrics such as CCCs. Additionally, while some of these responses (e.g., in S7) may reflect genuine cortical activity,

others (e.g., in S1–S3) could be due to peripheral effects such as muscle twitches, particularly when amplitudes are large. Determining their origin and functional significance will require future studies with tailored protocols.

While the active setup reduces preparation time, it is important to consider potential trade-offs in signal stability. In our experience, electrode impedance typically improves over the course of the recording due to progressive skin–electrode contact stabilization. Active electrodes, in particular, have consistently provided reliable signal quality throughout entire sessions. In this study, we did not observe any noticeable signal degradation or variability that would indicate a loss of contact quality or the need for electrode readjustment. Although impedance was not continuously re-evaluated during the session, it can be monitored periodically if needed. However, with increasing operator experience, real-time visual inspection of the EEG signal often proves sufficient to ensure signal integrity. We acknowledge that for longer protocols, continuous impedance monitoring may offer an additional safeguard, but for the present study, the setup proved robust over the typical session duration.

While our findings demonstrate the reliability and efficiency of the active setup for TMS–EEG recordings over M1, it is important to acknowledge that this region is among the more accessible and well-characterized stimulation targets. The generalizability of these results to other cortical areas, particularly more anterior or lateral regions such as the inferior frontal gyrus, remains to be established. These areas are often associated with increased artifact susceptibility due to anatomical and functional factors, including proximity to facial muscles and variations in skull curvature. As such, future work will be needed to systematically evaluate the performance of active electrodes in these more challenging contexts.

Finally, we acknowledge that our sample size of 10 participants may limit the generalizability of the findings. Future studies involving larger and more diverse cohorts – including clinical populations – are necessary to further validate and extend the conclusions presented here.

## Materials and methods
### Study participants
Ten healthy adults (9 male, 1 female), aged between 22 and 52 years (mean age: 36.8 years), voluntarily participated in this study. All participants had normal or corrected-to-normal vision and no history of neurological or psychiatric disorders. Prior to participation, each subject provided written informed consent in accordance with institutional and international ethical standards.

All experimental procedures adhered to the principles of the Declaration of Helsinki and were approved by the Ethics Committee of IRCCS Sicilia under approval number U148/19.

### Electrode types
In this study, we simultaneously recorded EEG using two different electrode types. As a gold-standard reference, we employed the widely used passive Multitrode Electrodes B18 for TMS (Easycap GmbH, Wörthsee, Germany), referred to as passive electrodes throughout the manuscript. As the second type, we used active g.LADYbird TMS electrodes (g.tec medical engineering GmbH, Schiedlberg, Austria), referred to hereafter as active electrodes.

Both electrode types feature an Ag/AgCl interface: a C-shaped intermittent ring for the passive electrodes and a cylindrical pellet for the active electrodes. The package height is both 3.5 mm for passive electrodes and 3.0 mm for active electrodes with a 0.5 mm cable junction protrusion on the latter. Both electrode types have a circular base shape, with diameters of 14 mm (passive) and 18 mm (active). Table 1 summarizes key specifications of the two electrode types, and Fig. 6 provides a technical drawing and photograph of the active electrodes.

### System overview
Figure 7 provides a schematic overview of the system setup used in this study, which included a TMS device, a biosignal amplifier with dedicated TMS-compatible electrode connector boxes, and an acquisition PC.

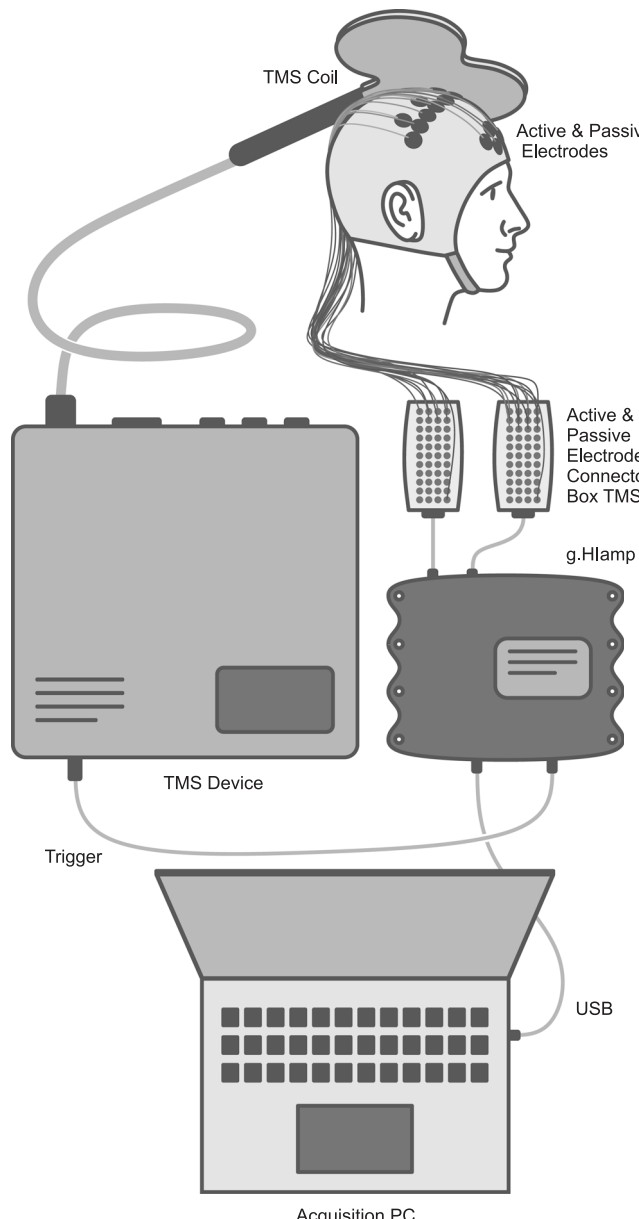

**Fig. 7 | Schematic system overview of the recording setup.** TMS Transcranial magnetic stimulation, USB Universal serial bus, PC Personal computer.

We used a PowerMAG Research stimulator (MAG & More GmbH, Munich, Germany) for TMS, capable of delivering high-frequency stimulation up to 15 Hz at full power with customizable pulse parameters. To ensure stable and precise positioning, we mounted the Double Coil PMD70 figure-of-eight stimulation coil (MAG & More GmbH) on a Manfrotto articulated arm with a Super Clamp and Avenger D200B grip head (Videndum Media Solutions, Cassola, Italy), secured to a tripod stand.

For EEG acquisition, we used a g.HIamp biosignal amplifier (g.tec medical engineering GmbH, Schiedlberg, Austria), which supports both active and passive electrodes. We connected the recording, ground, and reference electrodes to their respective active or passive electrode connector boxes (see Fig. 8 for the precise electrode placement). Notably, we used TMS-specific electrode connector boxes designed to suppress decay artifacts following magnetic pulses. To synchronize EEG recordings with TMS pulses, we routed the TTL trigger output from the TMS device to the digital trigger input of the g.HIamp.

We connected the g.HIamp to the acquisition PC via USB to enable real-time visualization, data storage, and subsequent signal processing.

### Electrode placement

To enable a performance comparison within and between electrode types, we designed the electrode placement according to three key criteria: (1) electrodes should be positioned as closely as possible while avoiding bridges, (2) active and passive electrodes should be interleaved, and (3) inter-electrode distances should be as uniform as possible. To satisfy these requirements, we selected a subset of the 10/05 electrode system, enabling precise and standardized positioning[48].

We arranged electrodes laterally across both motor cortices in an alternating pattern, starting at FCC5h over the left hemisphere and continuing to FCC6h on the right. In addition, we placed passive electrodes along the midline, spanning from C3 through Cz to C4. This configuration resulted in 11 passive electrodes positioned at FCC5h, FCC3h, FCC1h, FCC2h, FCC4h, FCC6h, C3, C1, Cz, C2, and C4, and 5 active electrodes located at FCC3, FCC1, FCCz, FCC2, and FCC4.

We placed ground and reference electrodes on the forehead, over the frontal lobe, for each electrode type: Fpz (active ground), Fp2 (passive ground), NFp2h (active reference), and AFp2 (passive reference). All electrodes were mounted on a g.GAMMAcap3 (g.tec medical engineering GmbH, Schiedlberg, Austria). The EEG cap was modified to support high-density 10/05 electrode placement by adding openings at the FCC3, FCC1, FCCz, FCC2, and FCC4 positions to ensure accurate electrode positioning. Figure 8 illustrates the topographical layout of all recording, ground, and reference electrodes.

### TMS–EEG recording

Before involving the subjects, we prepared the TMS–EEG setup as shown in Fig. 7, which included connecting all hardware components and electrodes, powering on the devices, and launching g.Recorder acquisition software (g.tec medical engineering GmbH, Schiedlberg, Austria) on the acquisition PC. For EEG acquisition, we set the sampling rate to 4.8 kHz, referenced active and passive channels to their respective reference electrodes, and recorded the reference electrodes relative to their respective ground channels.

We then seated the subject in a comfortable chair in a quiet room to minimize distractions and provide a stable recording environment. After cleaning the scalp at all recording sites with alcohol, we mounted the EEG cap with all electrodes pre-attached. Next, we prepared the passive electrodes by injecting conductive gel using a syringe and gently abrading the scalp under each electrode, while monitoring impedance in real time using g.Recorder. This preparation process took approximately 20–30 minutes for all 13 passive electrodes (recording, reference, and ground). Once impedance levels fell below 5 kΩ, we proceeded to prepare the active electrodes. These required no abrasion and achieved impedance levels below 20 kΩ with ease, allowing for a much faster preparation time of just 1–2 minutes for all 7 electrodes.

After successfully preparing all EEG electrodes, we set up the TMS coil. We targeted the left M1 by positioning the coil tangentially to the scalp over the C1 electrode location. The coil was oriented at approximately 45° to the sagittal plane, with the handle pointing posterior-laterally, thereby inducing an anterior-posterior posterior–anterior (AP-PA) current in the underlying cortex, as per standard procedures. We adjusted the coil position iteratively while delivering stimulation pulses, identifying the optimal motor hotspot by visually confirming motor-evoked thumb twitches in the contralateral (right) hand, caused by activation of the first dorsal interosseous (FDI) muscle. We did not use any auditory noise masking in order to maintain a naturalistic stimulation environment. Once the hotspot was located, we secured the coil mount and tripod to ensure consistent stimulation throughout the recording.

Following coil setup, we instructed the subject to fixate on a static point on a screen to minimize eye movements during the EEG recording. We recorded 30 seconds of resting-state EEG, confirming signal quality

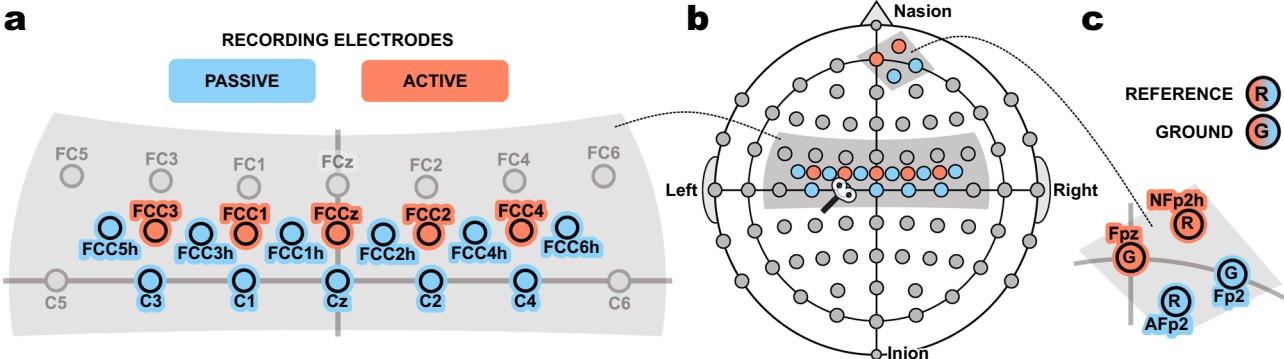

**Fig. 8 | Placement of electroencephalographic (EEG) electrodes for recording, reference, and ground. a** Close-up of recording electrode positions over the motor cortices based on the 10/05 system. **b** Overview of recording and ground/reference electrodes embedded in a 10/20 system. **c** Reference and ground electrodes located on the frontal lobe.

visually via g.Recorder's real-time scope. We then initiated TMS, delivering 100 biphasic pulses at a rate of 0.9 Hz and an intensity of 65%. After stimulation, we recorded another 30 seconds of resting-state EEG to again verify signal quality. We repeated this procedure 2–3 times, depending on the subject's comfort and well-being. All EEG data were saved for further analysis.

### Signal preprocessing

We processed the acquired EEG signals using MATLAB (The MathWorks, Inc., Natick, MA, USA) and g.BSanalyze (g.tec medical engineering GmbH, Schiedlberg, Austria) as follows:

1) **Re-referencing**. To ensure consistent signal conditions, we re-referenced all electrodes to the active reference electrode. The choice of active or passive electrodes does not affect the overall results or conclusions; we address this point in the Discussion.

2) **Trigger realignment**. Using the trigger signals recorded from the TMS device, we precisely identified the onset of each TMS pulse by detecting the corresponding stimulation pulse artifact onset in the EEG signal.

3) **Epoching**. Based on the realigned triggers, we epoched the data into trials from –0.1 seconds to +0.7 seconds relative to each TMS pulse onset.

4) **Trial rejection**. To remove trials contaminated by external artifacts (e.g., subject movement), we applied a statistical rejection procedure based on amplitude outliers. Since such artifacts are typically rare but exhibit large deviations, we proceeded as follows:

   a) Starting 1.5 ms after stimulation onset, we computed a modified $z$-score of the EEG samples $x[n, m, k]$ for each time point $n$, channel $m$, and trial $k$:

   $$z[n, m, k] = \frac{x[n, m, k] - \hat{\mu}[n, m]}{\hat{\sigma}[m]} \quad (1)$$

   Here, $\hat{\mu}[n, m]$, denotes the median across trials at time point $n$ and channel $m$, providing a robust estimate of the average response. The denominator $\hat{\sigma}[m]$ was computed as the square root of the time-averaged variance across trials, yielding a robust standard deviation estimate per channel.

   b) We defined an amplitude-related threshold $z_{thr} = 2.0$ and a sample-related threshold $N_{thr} = 0.3 \cdot f_s$, where $f_s$ is the sampling rate. This corresponds to 0.3 seconds suprathreshold activity. To simplify computation and account for both prolonged moderate deviations and shorter, stronger ones, we combined the thresholds into a single area-under-the-curve (AUC) threshold: $A_{thr} = z_{thr} \cdot N_{thr}$.

   c) For each channel and trial, we calculated the AUC of the modified $z$-scores. First, we took the absolute value of the $z$-scores and set all values below $z_{thr}$ to zero (ignoring normal fluctuations), then summed the remaining values across time points $n$, yielding a single AUC value per channel and trial.

   d) We rejected a trial if the AUC exceed $A_{thr}$ in any channel.

   Notably, we implemented this procedure in a strictly causal manner: starting after collecting 10 trials, we evaluated each subsequent trial individually, based solely on the statistics available up to that point. No retrospective re-evaluation or rejection of previously accepted trials was performed. Figure 1 documents the number of removed trials for each subject.

5) **Averaging**. We computed the arithmetic mean across the 100 TMS trials except the rejected ones.

6) **Decay artifact removal**. To remove the characteristic exponential decay artifact induced by the TMS pulse, we used the TMS artifact removal function implemented in g.BSanalyze. This algorithm models the post-stimulus decay artifact as a linear combination of multiple exponential decay functions and subtracts the resulting model from the raw signal (see Fig. 2 for examples). This procedure is applied to each channel and involves the following main steps:

   a) **Logarithmic resampling**. The signal is resampled using time points that are spaced logarithmically rather than linearly. This provides dense sampling in the early post-stimulus period (where the decay is fast) and sparser sampling in the later period (where the decay is slow), thus balancing the fitting accuracy across time scales.

   b) **Interval restriction**. Only samples later than 1.5 ms after TMS pulse onset are considered. Given a typical pulse duration of ≈1 ms, this ensures that fitting starts immediately after the end of stimulation.

   c) **Initial slope estimation**. The algorithm detects dominant deflections in the signal and estimates their polarity and characteristic decay time constant. These estimates help constrain the directionality of the modeled exponentials in the next step.

   d) **Template construction**. A dictionary of $N = 200$ exponential decay basis functions $v_i(t)$ is constructed, with time constants $\tau_i$ logarithmically spaced between 1 ms and 1 s:

   $$v_i(t) = \pm e^{-t/\tau_i} \text{ with } i = 1, \ldots, N \quad (2)$$

   The sign of each basis function is selected based on the previously estimated polarity of the decay components.

   e) **Non-negative least squares fitting**. The decay artifact is modeled as a weighted sum of these basis functions:

   $$w(t) = \sum_{i=1}^{N} c_i v_i(t) \quad (3)$$

   The coefficients $c_i \geq 0$ are estimated using a non-negative least squares (NNLS) fit, which ensures physically plausible, additive decay

components and avoids overfitting oscillatory, physiological responses.

 f) **Artifact subtraction**. Finally, the modeled artifact $w(t)$ is subtracted from the original EEG signal to yield the cleaned TEP response.

7) **Power-line interference cancellation**. Power-line noise was estimated and removed using a least-squares fit of the fundamental frequency (50 Hz) and its harmonics up to the Nyquist frequency. The fitted sinusoidal components were subtracted from the cleaned data obtained in the previous step.

8) **Post-hoc lowpass**. To attenuate higher frequency components, we finally applied a zero-phase boxcar filter of 3 ms length (corresponding to a cutoff frequency of ≈147.7 Hz).

These steps yielded clean TEPs at all electrode locations, which were then used for the signal consistency, amplitude variability, and convergence of averages analyses described below.

## Signal consistency analysis

In this study, we addressed two research questions related to signal consistency. First, we asked whether the novel active electrodes perform comparably to the passive gold standard: (1) Are TEPs consistent between active and passive electrodes? Second, we investigated the internal consistency of each electrode type: (2) Are TEPs consistent within active or within passive electrodes?

To answer these questions, we compared TEPs recorded from spatially neighboring electrodes. This approach relies on the assumption that closely spaced electrodes should capture nearly identical signals due to volume conduction, given that physiological variability at such small spatial scales is minimal or negligible. Therefore, any discrepancies in the recorded signals can be attributed primarily to electrode-specific factors, rather than underlying cortical dynamics.

We defined four comparison groups based on electrode proximity and type, as illustrated in Fig. 8 and detailed in Fig. 3a-d. Group A comprises 15 neighboring active–passive electrode pairs, each separated by ≈2.1 cm (Fig. 3a). This group directly addresses research question (1), evaluating signal consistency between electrode types. In contrast, Groups B through D address research question (2) by assessing within-type consistency: Group B consists of 10 diagonal passive–passive pairs with an inter-electrode distance of ≈3.0 cm (Fig. 3b). Group C includes 5 horizontal passive–passive pairs spaced ≈4.2 cm apart (Fig. 3c), and Group D consists of 4 horizontal active–active pairs at the same distance (Fig. 3d).

For each electrode pair in Groups A–D, we computed the Concordance Correlation Coefficient (CCC), a scale-sensitive variation of Pearson's correlation coefficient that simultaneously accounts for both correlation and agreement in signal amplitude. The CCC is defined as:

$$\rho_c = \frac{2\rho\sigma_x\sigma_y}{\sigma_x^2 + \sigma_y^2 + \left(\mu_x - \mu_y\right)^2} \tag{4}$$

where $x$ and $y$ denote the two TEP time courses being compared, $\mu_x$ and $\mu_y$ are their means, $\sigma_x^2$ and $\sigma_y^2$ are their variances, and $\rho$ the Pearson correlation coefficient. A CCC value of 1 indicates perfect agreement in both waveform shape and scale.

This analysis yielded a total of 150 CCC values for group A, 100 for group B, 50 for group C, and 40 for group D. We repeated the analysis for two distinct time windows, corresponding to early TEPs (15–80 ms) and late TEPs (80–350 ms), in line with commonly used temporal definitions in the literature[38,49].

## Amplitude variability analysis

A confounder of the signal consistency analysis is that TEPs vary even across closely spaced neighboring EEG electrodes. These variations are primarily related to overall signal amplitude rather than differences in waveform shape or morphology. To address this confounder, we aimed to identify which factors significantly contributed to the observed amplitude variability in the TEPs. Specifically, we assessed the effects of (1) distance from the stimulation site, (2) electrode type (active vs. passive), and (3) subject-specific variability.

We quantified each electrode's TEP amplitude using the root mean square (RMS) of early (15–80 ms) and late (80–350 ms) TEP components. To ensure statistical balance and improve interpretability, we restricted our analysis to the anterior electrode row (i.e., FCC5h, FCC3,…, FCC6h; see Fig. 8). To evaluate the contribution of the three candidate factors, we employed a linear mixed-effects model (LME) with TEP amplitude as the dependent variable. The model included distance (in centimeters) and electrode type (active vs. passive) as fixed effects. Given that each subject experienced all levels of both electrode type and distance (i.e., a fully within-subject design), we included subject-specific random slopes for both factors in addition to a random intercept. This allowed the model to account for inter-individual variability in baseline amplitude as well as in the effects of electrode type and distance. We fitted the model separately for early and late TEPs using the MATLAB fitlme function with 'RMS ~ type + distance + (1 + type + distance | subject)' as formula for model specification.

## Convergence of averages analysis

In a final analysis, we assessed the convergence of averaged TEPs as the number of included trials increased toward the full-trial average, computed from all non-rejected trials out of 100. To this end, we incrementally increased the number of collected trials from 10 to 100 in steps of 1. At each step, we re-applied the complete processing pipeline, yielding early and late TEPs for each subject and electrode channel. Note that, due to trial rejection, TEPs were not updated in steps where the current trial was excluded.

Using these stepwise averages, we computed the CCC between the current average and the full-trial average. We grouped the resulting CCCs by electrode type (active vs. passive) and TEP time window (early vs. late), allowing us to analyze convergence behavior separately for each condition.

## Ethics declarations

All experiments to collect the datasets followed the guidelines of the Declaration of Helsinki. The experimental procedure of the TMS–EEG study was approved by the Ethics Committee of IRCCS Sicilia under approval number U148/19.

## Reporting summary

Further information on research design is available in the Nature Portfolio Reporting Summary linked to this article.

## Data availability

Complete datasets collected within this study are available upon reasonable request. Please contact the corresponding author (gruenwald@gtec.at) for any inquiries.

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

## Acknowledgements
AZ was supported by the Italian Ministry of Health – Ricerca Corrente.

## Author contributions
J.G., L.S., S.S., M.B., A.Z., and C.G. conceptualized the study. J.G. and S.S. developed the methodology and implemented the software. J.G., L.S., S.S., A.B., G.L., A.C., M.F., and S.T. validated the results. JG and SS performed the formal analysis. L.S., A.B., G.L., A.C., and M.F. conducted the experiments. J.G. and S.S. curated the data. J.G., L.S., G.L., and A.C. wrote the original draft. S.S., M.P., R.S., M.B., A.Z., and C.G. reviewed and edited the manuscript. J.G., L.S., and S.S. prepared the figures and visualizations. M.P., R.S., M.B., and C.G. supervised the study. C.G. managed the project and acquired funding.

## Competing interests
The authors declare the following competing interests: JG, LS, SS, and ST were employed by g.tec medical engineering GmbH. CG served as the CEO of g.tec medical engineering GmbH. The remaining authors declare no competing interests.
