## [Transparent Peer Review file · Communications Engineering]

Reliable and Efficient Transcranial Magnetic Stimulation– Electroencephalography (TMS–EEG) Using Ultra-Thin Active Electrodes

Corresponding Author: Dr Johannes Gruenwald

Version 0:

Reviewer comments:

Reviewer #1

(Remarks to the Author)

Summary

The manuscript offers an insightful comparison between passive and novel ultra-thin active electrodes in the context of TMS-EEG, with a particular focus on their impact on signal quality and artifact reduction. Given that artifacts represent a major challenge in TMS-EEG research, the author's effort to systematically evaluate the potential of active electrodes is highly relevant.

The experimental design is logically sound, and the analyses are appropriate for addressing the central question. The authors convincingly demonstrate that ultra-thin active electrodes may be a reliable and efficient alternative to passive setups. This has important implications for both basic and clinical applications of TMS-EEG, where clean signal is critical.

Overall, this is a valuable contribution to the field, and I recommend it for publication following minor revisions:

Comment 1 - P. 10 (section "Active TMS-EEG is efficient"). The authors state that the proposed active setup also offers substantial gains in experimental efficiency, as preparation time was significantly reduced. While this is a noteworthy advantage, it may be important to consider potential trade-offs. Specifically, the manuscript does not address whether electrode impedance remains stable over the course of a recording session. In practice, a faster setup might lead to less durable contact quality, potentially requiring readjustments or leading to signal degradation over time (over the duration of the experimental session).

It would strengthen the manuscript if the authors could comment on whether impedance stability was monitored during the session (or re-evaluated at the end), or whether any variability in signal quality was observed that might suggest a need for re-evaluation of electrode placement mid-experiment. While the experimental session in this study may have been relatively short, which may limit the likelihood of differences, acknowledging this limitation would provide a more balanced perspective on the practical implications of adopting the active setup.

Comment 2 - The current study focuses exclusively on the primary motor cortex (M1), which is a common and practical choice for TMS-EEG validation. However, it is worth noting that this target site may not generalize to more challenging stimulation targets. In particular, lateral and anterior regions such as the inferior frontal gyrus (IFG) often present greater difficulties due to increased artifact susceptibility.

As such, while the results are promising for M1, the performance of active electrodes in more artifact-prone regions remains an open question. A brief discussion of this limitation (on P. 11) – and how it may influence the broader applicability of the setup – would add important context for researchers considering active electrodes for varied cortical targets.

Comment 3 - The authors state that a trial was rejected if the AUC (area-under-the-curve) exceed A_{thr} in any channel (P. 17, section "Signal Preprocessing"). For transparency, I would like to ask the authors to report how many trials, on average, were removed during preprocessing.

Comment 4 - The statistical approach used in the manuscript is modern and appropriate (P. 19, section "Amplitude Variability Analysis"). The authors use linear mixed models (LME) with electrode type and distance as fixed effects and model subject as a random intercept. Given that each subject experienced all levels of both electrode type and distance (i.e., a fully within-subject design), would it be appropriate to include random slopes for these factors as well? If supported by the data structure, this could help to account for inter-individual variability in how these factors affect the outcome measures, which may be particularly important given the relatively small sample size (as was acknowledged by the authors in the

limitations section). If random slopes were considered but not included (e.g., due to convergence issues), a brief rationale in the manuscript would enhance clarity and transparency.

Reviewer #2

(Remarks to the Author)

The primary objective of this study is to demonstrate that newly developed active electrodes can effectively record TMS-evoked responses that are comparable in quality to those obtained using conventional passive electrodes. A noteworthy advancement is that the proposed electrodes are significantly thinner than those currently available on the market, which may offer practical advantages in certain applications.

While the manuscript introduces promising developments, some key methodological details are lacking or insufficiently described.

Beginning with the abstract, the stated aims appear to be twofold: (1) to evaluate the performance of the new active electrodes, and (2) to implement "hardware- and software-based artifact suppression." The second aim is emphasized with the claim that a "fully automated algorithm" can reliably eliminate residual decay artifacts, thereby enabling rapid and standardized post-processing without expert intervention. However, the manuscript provides only a cursory explanation of this algorithm, and it remains unclear exactly what the automated pipeline entails. It appears that some components of the artifact correction may occur online, while others are applied post hoc, but this distinction is not clearly articulated. Overall my point is that there is not enough at the moment to justify claims on the analysis pipeline.

This lack of clarity continues in the discussion, where the authors state: "Our setup also mitigates the problem of strong or prolonged TMS decay artifacts." Yet, the manuscript does not present any figures or quantitative analyses demonstrating how the exponential fitting improves signal quality or reduces artifacts. Including such evidence would greatly strengthen the claim.

Additionally, the manuscript would benefit from a comparative visualization of signals recorded from active versus passive electrodes. A figure showing overlaid traces, either from electrodes near the stimulation site or based on RMS values, would help the reader assess the similarity in signal quality and better appreciate the performance of the new electrodes.

Version 1:

Reviewer comments:

Reviewer #1

(Remarks to the Author)

Thank you for the thoughtful and thorough revision. The changes have significantly improved the manuscript. I appreciate the time and effort the authors have dedicated to address the feedback, as well as the inclusion of the signals as early as 1.5 ms in the revised manuscript. I recommend acceptance of the current version of the manuscript and have no further comments.

Reviewer #2

(Remarks to the Author)

The authors clearly addressed my concerns and have implemented appropriate changes to improve the overall quality of the manuscript. In light of these changes, I can recommend the publication of the manuscript in its current version.

Reliable and Efficient TMS-EEG Using Ultra-Thin Active Electrodes

Response to the Reviewers

We sincerely thank the reviewers for their thoughtful and constructive comments. In the revised manuscript, we have addressed each point as comprehensively as possible.

During a final review of the dataset, we identified one aspect unrelated to the reviewer feedback that warranted revision. Specifically, our initial exclusion of data <10 ms post-stimulation was unnecessarily conservative. Upon reanalysis, we found that signals from as early as 1.5 ms after stimulation onset were intact and usable. Consequently, we reprocessed all data and updated the manuscript as follows:

- Adjusted the processing constraint from >10 ms to >1.5 ms [lines 430 and 468–470]
- Updated TEP time courses [Figure 1; lines 100–102; Supplementary Material]
- Minor updates in Signal Consistency results [lines 141ff; Figure 3; Supplementary Material]
- Minor updates in Amplitude Variability results [lines 165ff; Figure 4; Supplementary Material]
- Minor updates in Convergence of Averages results [Figure 5; Supplementary Material]
- Updated Discussion accordingly [lines 208 and 268–283]

Below, please find our point-by-point **response** to the reviewers. Line numbers refer to the version of the manuscript with Track Changes enabled.

Reviewer #1:

Comment 1 - P. 10 (section "Active TMS-EEG is efficient"). The authors state that the proposed active setup also offers substantial gains in experimental efficiency, as preparation time was significantly reduced. While this is a noteworthy advantage, it may be important to consider potential trade-offs. Specifically, the manuscript does not address whether electrode impedance remains stable over the course of a recording session. In practice, a faster setup might lead to less durable contact quality, potentially requiring readjustments or leading to signal degradation over time (over the duration of the experimental session).

It would strengthen the manuscript if the authors could comment on whether impedance stability was monitored during the session (or re-evaluated at the end), or whether any variability in signal quality was observed that might suggest a need for re-evaluation of electrode placement mid-experiment. While the experimental session in this study may have been relatively short, which may limit the likelihood of differences, acknowledging this limitation would provide a more balanced perspective on the practical implications of adopting the active setup.

Response: We thank the reviewer for this important observation. We added a corresponding paragraph to the Discussion addressing impedance stability, signal quality over time, and practical monitoring strategies [lines 290–300].

Comment 2 - The current study focuses exclusively on the primary motor cortex (M1), which is a common and practical choice for TMS-EEG validation. However, it is worth noting that this target site may not generalize to more challenging stimulation targets. In particular, lateral and anterior regions such as the inferior frontal gyrus (IFG) often present greater difficulties due to increased artifact susceptibility.

As such, while the results are promising for M1, the performance of active electrodes in more artifact-prone regions remains an open question. A brief discussion of this limitation (on P. 11) – and how it may influence the broader applicability of the setup – would add important context for researchers considering active electrodes for varied cortical targets.

Response: We agree and added a paragraph to the Discussion explicitly addressing this limitation and outlining directions for future studies targeting more challenging cortical regions [lines 301–308].

Comment 3 - The authors state that a trial was rejected if the AUC (area-under-the-curve) exceed Athr in any channel (P. 17, section "Signal Preprocessing"). For transparency, I would like to ask the authors to report how many trials, on average, were removed during preprocessing.

Response: The number of rejected trials is shown in Figure 1 on the right. To make this more transparent, we added a reference in the description of the trial rejection procedure [line 450].

Comment 4 - The statistical approach used in the manuscript is modern and appropriate (P. 19, section "Amplitude Variability Analysis"). The authors use linear mixed models (LME) with electrode type and distance as fixed effects and model subject as a random intercept. Given that each subject experienced all levels of both electrode type and distance (i.e., a fully within-subject design), would it be appropriate to include random slopes for these factors as well? If supported by the data structure, this could help to account for inter-individual variability in how these factors affect the outcome measures, which may be particularly important given the relatively small sample size (as was acknowledged by the authors in the limitations section). If random slopes were considered but not included (e.g., due to convergence issues), a brief rationale in the manuscript would enhance clarity and transparency.

Response: We appreciate this suggestion and have updated the model accordingly to include random slopes. The main findings remain unchanged. We revised both the Methods and Results sections to reflect this adjustment [lines 539–545 and 167–173].

Reviewer #2:

Beginning with the abstract, the stated aims appear to be twofold: (1) to evaluate the performance of the new active electrodes, and (2) to implement "hardware- and software-based artifact suppression." The second aim is emphasized with the claim that a "fully automated algorithm" can reliably eliminate residual decay artifacts, thereby enabling rapid and standardized post-processing without expert intervention. However, the manuscript provides only a cursory explanation of this algorithm, and it remains unclear exactly what the automated pipeline entails. It appears that some components of the artifact correction may occur online, while others are applied post hoc, but this distinction is not clearly articulated.

Overall my point is that there is not enough at the moment to justify claims on the analysis pipeline.

Response: We expanded the methodological details of the artifact removal procedure significantly [lines 453–486]. Additionally, we clarified the distinction between post-hoc application (as in our study) and real-time capabilities of the algorithm in the Discussion [lines 260–262]. To support the claims more directly, we added a dedicated Decay Artifact Removal subsection to the Results, including a new figure [Figure 2; lines 114–133; Supplementary Figure S2 for passive reference].

This lack of clarity continues in the discussion, where the authors state: "Our setup also mitigates the problem of strong or prolonged TMS decay artifacts." Yet, the manuscript does not present any figures or quantitative analyses demonstrating how the exponential fitting improves signal quality or reduces artifacts. Including such evidence would greatly strengthen the claim.

Response: As noted above, we added a Decay Artifact Removal subsection to the Results that explains the method and provides visual and quantitative evidence for its performance [Figure 2; lines 114–133; Supplementary Figure S2].

Additionally, the manuscript would benefit from a comparative visualization of signals recorded from active versus passive electrodes. A figure showing overlaid traces, either from electrodes near the stimulation site or based on RMS values, would help the reader assess the similarity in signal quality and better appreciate the performance of the new electrodes.

Response: While Figure 1 already provides a side-by-side comparison of active and passive electrode configurations, and Figure 3A includes a quantitative Signal Consistency comparison, we agree that a direct overlay of traces enhances interpretability. We therefore included such comparisons in the respective lower plots of Figure 2.